# Osteoporosis Is Associated with an Increased Risk of Colorectal Neoplasms Regardless of Sex: Nationwide Population-Based Cohort Study

**DOI:** 10.3390/diagnostics14060666

**Published:** 2024-03-21

**Authors:** Seung Heon Yoo, Ji Hyung Nam, Dong Jun Oh, Geun U Park, Sang Hoon Kim, Hyoun Woo Kang, Jae Hak Kim, Yun Jeong Lim

**Affiliations:** 1Department of Internal Medicine, Dongguk University Ilsan Hospital, Dongguk University College of Medicine, Goyang 10326, Republic of Korea; jnysh@naver.com (S.H.Y.); mileo31@naver.com (D.J.O.); drlimyj@gmail.com (Y.J.L.); 2GN Co., Seoul 05051, Republic of Korea; kokoko12305@naver.com; 3Department of Internal Medicine, Chung-Ang University Gwangmyeong Hospital, Chung-ang University College of Medicine, Gwangmyeong 14353, Republic of Korea; spring0107@naver.com; 4Department of Internal Medicine, Seoul Metropolitan Government Seoul National University Boramae Medical Center, Seoul National University College of Medicine, Seoul 07061, Republic of Korea; gangmali@naver.com; 5Department of Internal Medicine, Myongji Hospital, Goyang 10475, Republic of Korea

**Keywords:** bone mineral density, colorectal neoplasm, osteoporosis, sex

## Abstract

Vitamin D may have anticancer effects against colorectal cancer (CRC). Bone mineral density (BMD) reflects the long-term vitamin D status. This study investigated the association between osteoporosis and colorectal neoplasms (CRN). The data were obtained from the National Health Insurance Service sample cohort, which included 60,386 osteoporosis patients and 8224 controls who underwent BMD in 2002–2019. The logistic regression models included age, sex, income level, and comorbidity. Sensitivity tests were performed using the data from the National Health Screening Program. In total, 7706 (11.2%) patients were diagnosed with CRN, and the proportion was significantly higher in osteoporosis patients than in controls (11.7% vs. 8.1%). In the multivariate analysis, osteoporosis was associated with an increased risk of CRN (odds ratio (OR) = 1.91, 95% confidence interval = 1.75–2.09, *p* < 0.0001), which was significant for both colorectal adenomas and CRC (OR = 1.88 and 1.83, respectively). A subgroup analysis by sex revealed a significant association between osteoporosis and CRN in both women and men (OR = 2.06 and 1.66, respectively). The sensitivity tests revealed results similar to those of the original dataset. In conclusion, osteoporosis is significantly associated with CRN risk in both sexes. In high-risk patients with low BMD, appropriate screening for CRN and vitamin D supplementation are required, regardless of sex.

## 1. Introduction

Colorectal cancer (CRC) is a common cancer with a high mortality rate worldwide, leading to a global focus on its prevention and early detection [1,2]. Colorectal adenomas are precursors to CRC and arise from alterations in the normal mechanisms that regulate DNA repair and cell proliferation, known as the adenoma–carcinoma sequence [3,4]. In addition to the removal of colorectal adenomas and the early detection of CRC, identifying the risk and protective factors for colorectal neoplasms (CRN; including both colorectal adenoma and CRC) and addressing persons at risk are of utmost importance. Although various genetic and environmental factors are associated with the likelihood of developing CRC, several studies have reported that high intakes of vitamin D and calcium are associated with a reduced risk of CRC [5,6].

Osteoporosis commonly occurs in older adults. Increased age, menopause, smoking, decreased physical activity, and a lack of calcium and vitamin D intake are considered important risk factors for osteoporosis. Accordingly, appropriate calcium and vitamin D intake and regular exercise are important for achieving peak bone mass and preventing osteoporosis [7,8]. As the risk factors for osteoporosis are similar to those for CRC, vitamin D status is believed to play an important role in the significant correlation between osteoporosis and CRN. Recently, a cross-sectional study reported that osteoporosis increased the risk of colorectal adenomas in women aged >50 years, and another study found that the development of colorectal adenomas in postmenopausal women was inversely related to vitamin D levels [9,10].

However, most studies on the association between osteoporosis and the risk of CRN have been limited to elderly women or are single-center or multicenter cross-sectional studies that target patients at individual medical institutions [11,12]. In this study, we investigated the association between osteoporosis and CRN using claims data from the National Health Insurance Service (NHIS), which represents the entire population.

## 2. Materials and Methods

### 2.1. Study Population

This study used data from the NHIS Sample Cohort database, which is representative of the general South Korean population. Since 2002, the NHIS database has constructed an invaluable resource for patient data, including residence, income level, medical diagnoses, disability- or death-related data, and insurance claims. To investigate osteoporosis, we selected 175,512 individuals who had undergone bone mineral density (BMD) testing among the 1,137,861 individuals registered in the NHIS cohort during the period of 2002–2019. Patients aged <50 years, those who had already been diagnosed with CRC before the BMD test, and those with insufficient baseline information were excluded from the study (Figure 1). The diagnosis of colon cancer was established based on ICD-10 codes. Accordingly, all patients who were assigned a colon cancer diagnosis code even once before subject registration were excluded from the study. According to the World Health Organization diagnostic criteria, osteoporosis is defined as a BMD measured by dual-energy x-ray absorptiometry (DXA) of 2.5 standard deviations below than the reference mean value for young adults (a T-score of ≤−2.5) [13]. The case group (osteoporosis patients) consisted of those who had the M81 and M80 codes of the International Classification of Diseases (ICD-10) of osteoporosis and had a record of being prescribed treatment for osteoporosis during the study period according to the following Anatomical Therapeutic Chemical codes, M05BA04 (alendronic acid), M05BB03 (alendronic acid and cholecalciferol), M05BA07 (risedronic acid), M05BB07 (risedronic acid and cholecalciferol), M05BA06 (ibandronic acid), M05BA08 (zoledronic acid), M05BX04 (denosumab), G03XC01 (raloxifene), and H05AA02 (teriparatide). The control group consisted of individuals who did not have an ICD-10 code for osteoporosis and had never received a prescription for osteoporosis after excluding 51,038 subjects who met the exclusion criteria among the 175,512 individuals who underwent bone mineral density (BMD) testing. This study was approved by the Institutional Review Board of the Dongguk University Ilsan Hospital (IRB no. DUIH 2021-07-021). Because this study analyzed the NHIS secondary data for research and statistical purposes, the requirement for informed consent was waived by the IRB.

### 2.2. Study Variables

The main outcome was the diagnosis of CRN after the BMD test. Based on the ICD-10 codes, CRN included tubular, serrated, or villous adenomas (D12.0, D12.2–12.8), adenoma with high-grade dysplasia or carcinoma in situ (D01.0–01.2), and CRC (C18–20).

This study was designed to compare the incidence of CRN between the case and control groups. Several characteristics recorded in the NHIS cohort were compared. Age was divided into seven groups: 50–54, 55–59, 60–64, 65–69, 70–74, 75–79, and ≥80 years. All citizens of South Korea are obliged to subscribe to the National Health Insurance and pay insurance premiums based on their salary and property. The income levels were determined by the health insurance premium decile of the claims data (0–3, 4–7, and 8–10; higher scores indicate higher levels). The Charlson Comorbidity Index (CCI) was calculated by scoring comorbid conditions such as cardiovascular disease, diabetes, chronic lung disease, liver disease, kidney disease, and cancer, which are highly correlated with patient death. In this study, CCI was classified into four groups according to the score: 0 (low risk), 1, 2, and 3 or more (high risk).

### 2.3. Statistical Analyses

A chi-squared test was performed to compare the baseline characteristics and occurrence of CRN between osteoporosis patients and controls. Univariate and multivariate logistic regression analyses were used to estimate the odds ratios (ORs) and 95% confidence intervals (CIs) to determine the association between osteoporosis and CRN, followed by sub-analyses according to histological classification (low-grade adenoma, high-grade adenoma/carcinoma in situ, and invasive CRC). The models included age, sex, income, and the CCI score. We also performed subgroup analyses according to sex, age, income, and CCI scores to determine how the ORs from the regression model differed according to baseline characteristics. Moreover, to check the reliability of the national data, sensitivity analyses were performed using the sub-data set included in the national health screening program (NHSP). Statistical significance was set at *p* value less than 0.05. All statistical analyses were performed using SAS 9.4 version (SAS Institute, Cary, NC, USA).

## 3. Results

### 3.1. Baseline Characteristics

Overall, 68,610 individuals (60,386 patients and 8224 controls) were included in this study. Table 1 summarizes the baseline clinical characteristics of the study groups. A total of 7706 (11.2%) patients were diagnosed with CRN, and this proportion was significantly higher in the osteoporosis group than in the control group (11.7% vs. 8.1%, *p* < 0.0001). Significant differences were observed in the subclasses of colorectal adenomas and CRC. In a comparison of the other variables, the proportion of women was higher in the osteoporosis group than in the control group (89.8% vs. 52.2%). Osteoporosis tended to become more common with increasing age (*p* < 0.0001). The CCI score was slightly lower in the osteoporosis group. The osteoporosis patients and the control group did not differ significantly in terms of income levels.

### 3.2. Risks for Colonic Neoplasms in Osteoporosis; Analyses by Histologic Subclasses

Table 2 shows the univariate and multivariate logistic regression analyses of the association between osteoporosis and CRN according to the histological subclasses. The CRN rates were significantly higher in the osteoporosis patients than in the controls (OR = 1.49, 95% CI = 1.37–1.62, *p* < 0.0001). In terms of histologic subclass, both non-invasive the CRN, including all adenomas and carcinoma in situ (CIS) (OR = 1.38, 95% CI = 1.27–1.51, *p* < 0.0001), and the invasive CRC (OR = 1.90, 95% CI = 1.57–2.30, *p* < 0.0001) showed a significant increase with osteoporosis. In the multivariate analysis adjusted by other covariates (sex, age, CCI, and income level), osteoporosis was associated with an increased risk of CRN (OR = 1.91, 95% CI = 1.75–2.09, *p* < 0.0001), which was also significant in terms of both low-grade adenoma and invasive CRC (OR = 1.89 & OR = 1.83, both *p* < 0.0001). Noninvasive CRN was associated with osteoporosis (OR = 1.88, *p* < 0.0001); high-grade adenoma/CIS alone did not show an association with osteoporosis, which was related to the small sample size.

### 3.3. Subgroup Analyses by Baseline Characteristics

Next, we performed a subgroup analysis by age, sex, CCI, and income level to determine an association between osteoporosis and CRN, despite the differences in baseline characteristics between the subgroups (Table 3). The association between osteoporosis and CRN was statistically significant in all the age groups except those aged 80 years or older (OR = 1.30, 95% CI = 0.78–2.16, *p* = 0.3202). Osteoporosis was also a risk factor for CRN in both sexes, regardless of CCI and income levels. The risk of osteoporosis-related CRN was higher in women than in men (OR = 2.06 vs. 1.66, both *p* < 0.0001). The ORs tended to increase as the income levels increased.

### 3.4. Sensitivity Analysis

Of the 68,610 study subjects, sensitivity tests were performed on only 35,099 individuals who had participated in the NHSP for cancer screening (30,490 osteoporosis patients and 4609 controls). The baseline characteristics are shown in Appendix A. In the univariate analysis, CRN had a significant correlation with osteoporosis (OR = 1.37, 95% CI = 1.23–1.53, *p* < 0.0001) (Table 4). After adjustment for sex, age, CCI, income level, body mass index (BMI), and smoking and drinking habits, significant results were also observed (OR = 1.75, 95% CI = 1.56–1.97, *p* < 0.0001). In the analysis of each histological subclass, low-grade adenoma, noninvasive CRN, and invasive CRC were all significantly associated with osteoporosis, which was consistent with the main outcomes of the study.

## 4. Discussion

This nationwide, retrospective cohort study used claims data from a sample cohort from the NHIS to investigate the relationship between osteoporosis and CRN. Osteoporosis was significantly correlated with the risk of CRN, noninvasive CRN, and invasive CRC. In the subgroup analyses according to baseline characteristics, osteoporosis increased the risk of CRN in most age groups, regardless of sex, CCI, or income level. The main results of this study are consistent with the sensitivity analyses using the NHSP data.

To the best of our knowledge, this was the first study to demonstrate on a nationwide scale that osteoporosis is associated with an increased risk of CRN, regardless of sex. Taiwan and Denmark also published national cohort studies in 2012; however, all were limited to women taking oral bisphosphonates [14,15]. In the latter, the incidence and mortality rates for CRC were significantly lower in women taking oral bisphosphonates. Although the study did not assess BMD, it can be assumed that increased bone density is associated with CRC prevention, which is consistent with our findings. A multicenter study in Korea confirmed that osteoporosis is an independent risk factor for colorectal and high-risk adenomas. However, in the subgroup analysis according to sex, osteoporosis increased the risk of adenoma only in women (OR = 1.66, *p* = 0.034 vs. OR = 1.45, *p* = 0.417), which is believed to be due to the limited sample size [12]. Our study targeted a larger sample size and found that osteoporosis increased the risk of CRN regardless of sex, although the OR for this risk was higher in women than in men (OR = 2.06 vs. OR = 1.66, both *p* < 0.0001). This statistical difference provides some support for the previous multicenter study, but it may also be due to higher statistical power in women from a larger sample.

Epidemiologic studies have suggested that lower BMD is associated with an increased risk for colorectal adenoma/cancer, especially in postmenopausal women [9,11,12]. The biological mechanisms linking bone mass to the risk of CRC are not fully understood; however, cumulative exposure to estrogen may play a role, and so may vitamin D deficiency. Vitamin D is a steroidal hormone that plays a major role in the regulation of bone metabolism. Calcitriol (1.25(OH)_2_D) is an active form of vitamin D that is considered to exert anticancer effects through its function as a transcription factor and a regulator of differentiation, as well as immune response. Several studies have reported that vitamin D deficiency increases CRC incidence [16,17,18,19]. A prospective cohort study showed that vitamin D intake was associated with a reduced risk of early-onset CRC before the age of 50 years and CRC precursors (i.e., conventional adenomas and serrated polyps) [20]. Furthermore, a meta-analysis of case–control studies suggested an inverse association between serum 25(OH)D levels, the circulating form of vitamin D in the body, and colorectal adenomas in both Western and Asian populations [21]. Thus, adequate vitamin D supplementation is important for the prevention of CRC.

The biologically active form of vitamin D, calcitriol (1.25(OH)_2_D), binds to the vitamin D3 receptor (VDR). The VDR–RXR heterodimers, in which VDR and retinoid X receptor (RXR) are combined, bind to the vitamin D response element (VDRE) and activate or inhibit gene expression to produce vitamin D anti-neoplastic activity [17]. In colorectal cancer, 1.25(OH)_2_D exerts antitumor effects by suppressing cell proliferation, sensitizing apoptosis, and inhibiting angiogenesis [22]. Vitamin D is primarily produced upon exposure to sunlight. Therefore, the circulating 25(OH)D concentration is lower in winter than in summer, with less sun exposure, and can even change depending on the day, night, and next morning [23,24]. Moreover, medications such as antiepileptic drugs, bile acid sequestrants, metformin, and glucocorticoids can affect circulating 25(OH)D [25,26]. Bone mineral density changes throughout life, increases rapidly during adolescence, plateaus in the 30s, and then gradually decreases with age. In addition, BMD can change due to pregnancy, lactation, or medications that affect bone metabolism (e.g., contraceptives, glucocorticoids, and antiepileptic drugs), but it occurs slowly, over several months to years, compared to serum 25(OH)D concentration [27].

Because serum 25(OH)D has a short half-life of several weeks, it may not be indicative of an individual’s overall vitamin D status. A randomized controlled trial confirmed that there was a dose effect on 25(OH)D concentration when comparing the monthly intake of vitamin D after 12 months, but the study concluded that there was no between-group difference in BMD [28]. However, another double-blind, randomized controlled trial demonstrated that the BMD significantly increased after 15 and 30 months of continuous administration of calcium and vitamin D compared to that in the placebo group [29]. At this point, BMD can be considered a reflection of long-term vitamin D status over months to years, and is used as a more reliable biomarker to confirm the anti-neoplastic effect on CRN. Of course, if medication data were available, the vitamin D hypothesis could be directly addressed instead of being indirectly hypothesized via osteoporosis, and the correlation between BMD and vitamin D levels could also be assessed. It is unfortunate that the vitamin D levels of osteoporosis patients and controls could not be compared in this study. It would be worth including this assessment in subsequent studies.

In this study, we used claims data extracted from 1,137,861 cohorts registered in the NHIS database. Most Koreans (97%) are enrolled in the NHIS program, and the greatest advantage of these data is that they cover almost the entire population and are the closest to real-world data [30]. Consequently, the claims data used in this study are considered reliable and statistically significant.

Our study had several limitations. First, the secondary data we used had not been collected for specific research purposes. The omission of any important covariates might have resulted in residual confounding [31]. Therefore, when organizing the case and control groups, we checked not only whether osteoporosis was diagnosed, but also whether osteoporosis medication was taken during the study period. Moreover, considering that the sensitivity analysis was consistent with the main results of the study, the limitations of the secondary data were overcome to some extent. Second, the baseline characteristics of the case and control groups differed. In the control group, the male-to-female ratio was relatively even (male: female = 47.8% vs. 52.2%); however, in the case group, there were more females (M: F = 10.2% vs. 89.8%). The patients with osteoporosis were older than the control group. This difference can naturally occur when osteoporosis is common in both women and elderly individuals. To overcome this difference, we conducted subgroup analyses for sex, age, CCI, and income level as baseline characteristics, and most of these factors provided significant results. Finally, because only those aged 50 years or older who underwent BMD were included, the number of osteoporosis patients was much higher than that in the control group. In this regard, a selection bias might have occurred, but it was considered negligible because this was a nationwide database that did not target patients in hospitals.

## 5. Conclusions

Osteoporosis is a risk factor for CRN regardless of sex. Although vitamin D supplements or bisphosphonates may improve bone density, the results of this study do not directly prove the effectiveness of the use of supplements or medications for CRN. However, this study shows that improving bone density is associated with a decrease in CRN and provides indirect evidence for the need for vitamin D supplementation. The advantage of the study is that it was conducted on a large nationwide scale, which has not been reported before. Accordingly, appropriate screening for CRN is required in high-risk groups with low BMD, and sufficient vitamin D intake is recommended. Further large-scale prospective studies are needed to confirm our findings.

## Figures and Tables

**Figure 1 diagnostics-14-00666-f001:**
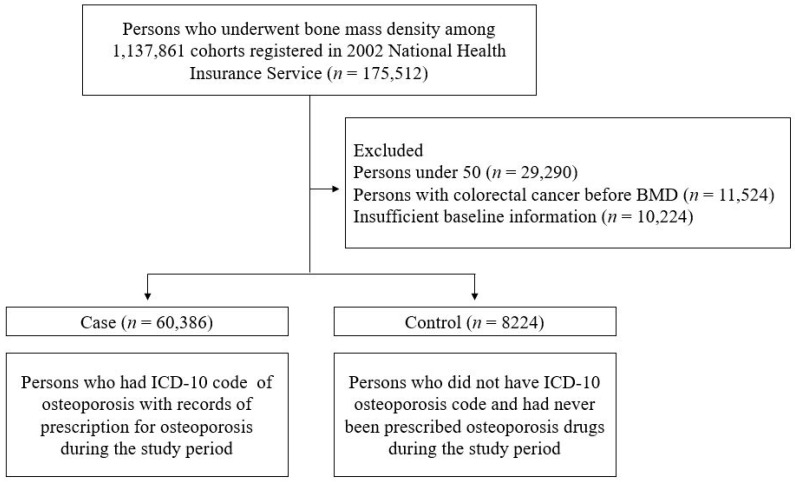
Schemes of the study.

**Table 1 diagnostics-14-00666-t001:** Baseline characteristics of the study population.

Variables	Total, *n* (%)	Osteoporosis, *n* (%)		* p * Value
	*n* = 68,610	No (*n* = 8224)	Yes (*n* = 60,386)	
Colorectal neoplasms				<0.0001
No	60,904 (88.8)	7556 (91.9)	53,348 (88.3)	
Yes	7706 (11.2)	668 (8.1)	7038 (11.7)	
Low-grade adenoma				<0.0001
No	62,263 (90.7)	7644 (92.9)	54,519 (90.4)	
Yes	6347 (9.3)	580 (7.1)	5767 (9.6)	
High-grade adenoma/CIS				0.6563
No	68,458 (99.8)	8204 (99.8)	60,254 (99.8)	
Yes	152 (0.2)	20 (0.2)	132 (0.2)	
Non-invasive CRN *				<0.0001
No	62,139 (90.6)	7630 (92.8)	54,509 (90.3)	
Yes	6471 (9.4)	594 (7.2)	5877 (9.7)	
Invasive CRC				<0.0001
No	66,908 (97.5)	8109 (98.6)	58,799 (97.4)	
Yes	1702 (2.5)	115 (1.4)	1587 (2.6)	
Sex				<0.0001
Male	10,076 (14.7)	3931 (47.8)	6145 (10.2)	
Female	58,534 (85.3)	4293 (52.2)	54,241 (89.8)	
Age, years				<0.0001
50–54	8854 (12.9)	2251 (27.4)	6603 (10.9)	
55–59	10,504 (15.3)	1576 (19.2)	8928 (14.8)	
60–64	11,691 (17.0)	1155 (14.0)	10,536 (17.5)	
65–69	14,317 (20.9)	1016 (12.4)	13,301 (22.0)	
70–74	11,028 (16.1)	987 (12.0)	10,041 (16.6)	
75–79	6863 (10.0)	660 (8.0)	6203 (10.3)	
≥80	5353 (7.8)	579 (7.0)	4774 (7.9)	
CCI				<0.0001
0	16,663 (24.3)	1839 (22.4)	14,824 (24.6)	
1	17,694 (25.8)	1795 (21.8)	15,899 (26.3)	
2	13,936 (20.3)	1588 (19.3)	12,348 (20.4)	
3 or more	20,317 (29.6)	3002 (36.5)	17,315 (28.7)	
Income level				0.7050
0–3	16,196 (23.6)	1961 (23.8)	14,235 (23.6)	
4–7	22,909 (33.4)	2714 (33.0)	20,195 (33.4)	
8–10	29,505 (43.0)	3549 (43.2)	25,956 (43.0)	

* Noninvasive CRN included all colorectal adenomas and carcinoma in situ. CCI, Charlson comorbidity index; CIS, carcinoma in situ; CRC, colorectal cancer; CRN, colorectal neoplasm.

**Table 2 diagnostics-14-00666-t002:** Univariate and multivariate logistic regression analyses for the association between osteoporosis and colonic neoplasms: subgroup by histology.

Variables	Osteoporosis, *n* (%)		Crude OR (95% CI)	*p* Value	Adjusted OR (95% CI)	*p* Value
	No	Yes				
Colorectal neoplasms	668	7038	1.49 (1.37–1.62)	<0.0001	1.91 (1.75–2.09)	<0.0001
Low-grade adenoma	580	5767	1.39 (1.27–1.52)	<0.0001	1.89 (1.72–2.09)	<0.0001
High-grade adenoma/CIS	20	132	0.90 (0.56–1.44)	0.6564	1.30 (0.77–2.19)	0.3190
Non-invasive CRN *	594	5877	1.38 (1.27–1.51)	<0.0001	1.88 (1.71–2.07)	<0.0001
Invasive CRC	115	1587	1.90 (1.57–2.30)	<0.0001	1.83 (1.49–2.24)	<0.0001

* Noninvasive CRN included all colorectal adenomas and carcinoma in situ. Other covariates (sex, age, Charlson Comorbidity Index score, and income level) were also adjusted. CIS, carcinoma in situ; CRC, colorectal cancer; CRN, colorectal neoplasm.

**Table 3 diagnostics-14-00666-t003:** Multivariate logistic regression analysis for the association between osteoporosis and colonic neoplasms: subgroups by baseline characteristics.

Variables	Adjusted OR	95% CI	*p* Value
Sex			
Male	1.66	1.45–1.91	<0.0001
Female	2.06	1.83–2.32	<0.0001
Age			
50–54	1.95	1.66–2.30	<0.0001
55–59	1.85	1.53–2.24	<0.0001
60–64	2.07	1.64–2.60	<0.0001
65–69	1.59	1.25–2.02	0.0001
70–74	2.28	1.69–3.07	<0.0001
75–79	1.62	1.10–2.40	0.0148
≥80	1.30	0.78–2.16	0.3202
CCI			
0	1.80	1.48–2.20	<0.0001
1	1.88	1.55–2.27	<0.0001
2	1.79	1.48–2.17	<0.0001
3 or more	2.04	1.76–2.38	<0.0001
Income level			
0–3	1.69	1.40–2.02	<0.0001
4–7	1.71	1.46–1.99	<0.0001
8–10	2.23	1.94–2.57	<0.0001

Other covariates (sex, age, Charlson Comorbidity Index score, and income level) were also adjusted.

**Table 4 diagnostics-14-00666-t004:** Sensitivity analysis using data from the National Health Screening Program (*n* = 35,099).

Variables	Osteoporosis		Crude OR (95% CI)	*p* Value	Adjusted OR (95% CI)	*p* Value
	No	Yes				
Colonic neoplasms	419	3679	1.37 (1.23–1.53)	<0.0001	1.75 (1.56–1.97)	<0.0001
Low-grade adenoma	378	3161	1.29 (1.16–1.45)	<0.0001	1.74 (1.54–1.96)	<0.0001
High-grade adenoma/CIS	14	70	0.75 (0.42–1.34)	0.3375	1.26 (0.65–2.41)	0.4931
Non-invasive CRN *	387	3211	1.28 (1.15–1.43)	<0.0001	1.73 (1.53–1.95)	<0.0001
Invasive CRC	54	672	1.90 (1.44–2.51)	<0.0001	1.78 (1.32–2.39)	<0.0001

* Noninvasive CRN included all colorectal adenomas and carcinoma in situ. Other covariates (sex, age, Charlson Comorbidity Index, income level, BMI, smoking, and alcohol consumption) were adjusted. CIS, carcinoma in situ; CRC, colorectal cancer; CRN, colorectal neoplasm.

## Data Availability

Data are contained within the article.

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
