# Peer review of "Osteoporosis Is Associated with an Increased Risk of Colorectal Neoplasms Regardless of Sex: Nationwide Population-Based Cohort Study"

_diagnostics, 2024, doi:10.3390/diagnostics14060666_

Round 1

Reviewer 1 Report

Comments and Suggestions for Authors

The paper „Osteoporosis is associated with an increased risk of colorectal neoplasms regardless of sex: nationwide population-based cohort study” present an interesting correlation beetwen two diseases with a constant increase in last decade. Epidemiologic studies suggest that lower bone mineral density (BMD) is associated with an increased risk for colorectal adenoma/cancer, especially in postmenopausal women. The biologic mechanisms linking bone mass to the risk of colon cancer are not fully understood; however, cumulative exposure to estrogen may play a role and also the vitamin D deficiency. Vitamin D is considered capable to 200 exert anticancer effects through its function as a transcription factor, regulator of 201 differentiation, and immune response.  Several studies demonstrated that vitamin D 202 deficiency increases CRC incidence like a prospective cohort study that showed that 203 vitamin D intake was associated with a reduced risk of early-onset CRC and CRC precursors (i.e., conventional adenomas and serrated polyps). Hormonal influences in female colorectal cancer patients may play a role in the increased risk of osteoporosis. In other cancers, ovarian dysfunction and early menopause increases the risk of osteoporosis.

It is important to specify whether the diagnosis of colorectal cancer was established at the time when you registered the case or if it is a survivor and if yes, how many years after the diagnosis of CRC was the osteoporosis assessment carried out.

Statistical analyses is very complex and the results well organised. If you have the data on the studied population, it would be interesting to statistically evaluate if the group of patients with osteoporosis and cancer had significant changes in the level of vitamin D. If you do not have this data for the studied batches, it would be worth including this evaluation in another paper. 

In the discussion chapter, please analyze in more detail the results of the study in relation to the data previously presented in the literature, even if there are few .

Reviewer 2 Report

Comments and Suggestions for Authors

Yoon and colleagues report an association of osteoporosis and colorectal neoplasia based on public health data from South Korea.  All patients who had received osteodensitometry and appropriate medication, including vitamin D, were compared with a control cohort without these criteria. The patients with osteoporosis had a significantly higher incidence of colorectal neoplasia.  The authors hypothesise that this is mediated by a long-term effect of vitamin D. 

Was the control cohort age and sex-matched, or how exactly did the authors arrive at these 8224 individuals? 

Is there an independent confirmatory cohort or is it possible to generate one?

If the medication data are available and vitamin D supplementation is given, the authors cannot directly prove or disprove the effect of vitamin D supplementation on the incidence of colorectal adenomas. 

Comments on the Quality of English Language

Methods: 

I am unable to trace the size and composition of the control cohort in this paper; this should be better explained and justified in the methods. 

Discussion: 

Since medication data are available, the vitamin D hypothesis could be directly addressed instead of being indirectly hypothesised via osteoporosis. This should at least be attempted.

Round 2

Reviewer 2 Report

Comments and Suggestions for Authors

The questions have been adequately addressed

Comments on the Quality of English Language

English can be improved for example: 

line 268:

Moreover, age was generally  higher in the case group than in the control group.

could be changed to: 

The patients with osteoporosis were older than the control group

line 282 :  This study was  meaningful because it was conducted on a large nationwide scale, which has not been 283 previously reported

should be changed to 

The advantage of the study is that it was conducted on a large nationwide scale, which has not been reported before